

# Evaluation of Monte Carlo tools for high energy atmospheric physics

Casper Rutjes[1], David Sarria[2], Alexander Broberg Skeltved[3], Alejandro Luque[4], Gabriel Diniz[5,6], Nikolai Østgaard[3], and Ute Ebert[1,7]

[1]Centrum Wiskunde & Informatica (CWI), Amsterdam, The Netherlands
[2]Astroparticules et Cosmologie, University Paris VII Diderot, CNRS/IN2P3, France
[3]Department of Physics and Technology, University of Bergen, 5020 Bergen, Norway
[4]Instituto de Astrofisica de Andalucia (IAA-CSIC), PO Box 3004, Granada, Spain
[5]Instituto Nacional de Pesquisas Espaciais, Brazil
[6]Instituto de Física, Universidade de Brasília, Brazil
[7]Eindhoven University of Technology, Eindhoven, The Netherlands

*Correspondence to:* Casper Rutjes (casper.rutjes@cwi.nl)

**Abstract.** The emerging field of high energy atmospheric physics (HEAP) includes Terrestrial Gamma-ray Flashes, electron-positron beams and gamma-ray glows from thunderstorms. Similar emissions of high energy particles occur in pulsed high voltage discharges. Understanding these phenomena requires appropriate models for the interaction of electrons, positrons and photons of up to 40 MeV energy with atmospheric air. In this paper we benchmark the performance of the Monte Carlo codes

5 Geant4, EGS5 and FLUKA developed in other fields of physics and of the custom made codes GRRR and MC-PEPTITA against each other within the parameter regime relevant for high energy atmospheric physics. We focus on basic tests, namely on the evolution of monoenergetic and directed beams of electrons, positrons and photons with kinetic energies between 100 keV and 40 MeV through homogeneous air in the absence of electric and magnetic fields, using a low energy cut-off of 50 keV. We discuss important differences between the results of the different codes and provide plausible explanations. We also

10 test the computational performance of the codes. The supplementary material contains all results, providing a first benchmark for present and future custom made codes that are more flexible in including electrodynamic interactions.

## 1 Introduction

### 1.1 Phenomena in high energy atmospheric physics

Thunderstorms have been observed to produce Terrestrial Gamma-ray Flashes (TGFs) (Fishman et al., 1994) and electron-

15 positron beams (Dwyer et al., 2008b; Briggs et al., 2011). Also long-lasting x-ray and gamma-ray glows have been observed near thunderclouds (McCarthy and Parks, 1985; Eack et al., 1996).

Two possible theories are currently under discussion, as reviewed by Dwyer et al. (2012), to create these phenomena by run-away electrons (Wilson, 1924), which may further grow in the form of so called relativistic run-away electron avalanches (RREA), introduced by Gurevich et al. (1992).



The first theory has been called the Cold Runaway theory (Gurevich, 1961) where thermal electrons are accelerated into the run-away regime within the strong electric fields of a transient discharge. Theoretical literature first focussed on the phase of the streamer discharge (Moss et al., 2006; Li et al., 2009; Chanrion and Neubert, 2010), and later on leader discharges (Celestin and Pasko, 2011; Celestin et al., 2012; Chanrion et al., 2014). Cold runaway is certainly at work in high energy emissions from nanosecond pulsed discharges (Stankevich and Kalinin, 1967; Kostyrya et al., 2006; Tarasenko et al., 2008; Shao et al., 2011) and during the formation of long sparks (Noggle et al., 1968; Dwyer et al., 2008a; Rep'ev and Repin, 2008; Cooray et al., 2009; Kochkin et al., 2012, 2014) in high voltage and pulsed plasma technology.

The second theory is the Relativistic Feedback Discharge model by Dwyer (2003). It is based on sustaining the RREA multiplication of the relativistic electrons in sufficiently high electric fields within a thunderstorm, by feedback of photons and positrons creating new avalanches (Babich et al., 2005; Dwyer, 2007, 2012). The first electrons are typically supplied by cosmic particles from the sun or from other galactic or extragalactic sources.

An extreme case both of cold or of RREA would be a Relativistic Runaway Electron Front where the density of runaway electrons is high enough to provide electric screening behind the ionization front (Luque, 2014).

We remark as well that a sufficiently energetic cosmic particle can create an extensive air shower with very high electron density in the shower core even in the absence of any electric fields; such densities were used by Dubinova et al. (2015) to explain lightning inception; and these air showers were also used to measure electric fields in thunderstorms (Schellart et al., 2015; Trinh et al., 2016). Radioactive decay is another source of high energy particles in the atmosphere.

All these phenomena require tracing the propagation of energetic electrons, photons, and also positrons through air, as well as modeling their interaction with air molecules and the subsequent scattering and energy loss or even total loss of the primary particles, together with the generation of secondary particles.

## 1.2 The multiple scales in energy and length

There are two basic problems for simulating these high energy phenomena in our atmosphere, related to the wide range of scales in energy and length.

First, the models have to bridge energy scales from tens of MeV down to thermal energies of tens of meV ($300\,\mathrm{K} \rightarrow 0.03\,\mathrm{eV}$), i.e., over 9 orders of magnitude. At the upper edge of this energy range, models developed by the high energy physics community (e.g., for CERN) exist where it should be noted that they were originally developed for even higher particle energies, and for the interaction of energetic particles with metals rather than with air — though radiation medicine now also develops models for the penetration of energetic radiation into biological tissue (Andreo, 1991; Sempau et al., 2001; Carrier et al., 2004), which consists mostly of similarly light molecules as air, but in the liquid rather than the gaseous state. In the low energy regime, models by the low temperature plasma physics community should be used, with cross-sections listed, e.g., on the community webpage (Pancheshnyi et al., 2012).

Second, in particular, for cold run-away models, there are two widely separated spatial scales: the source region with high and time dependent self-consistent electrodynamic fields where electrons are accelerated, and the wide propagation distances from the source to detectors in space or on ground where electric fields can be neglected.



**Table 1.** Codes used in this benchmark, their validity range (usable energy interval) and relative performance (normalized to the fastest code), possible inclusion of electric and magnetic fields (**E & B**) and self-consistent fields due to space charge. It should be noted that the synchronous particle tracking in GRRR, for the possible inclusion electric fields due to space charge, and the simulation without low energy cut-off approximation in MCPEP limits their performance. See for more descriptions Sect. 3.

| Code | Validity range (eV) | Relative perform. | E & B | Space charge |
|------|---------------------|-------------------|-------|--------------|
| EGS5 | $[10^4, 10^{11}]^a$ | 4.02 | N & N | N |
| FLUKA | $[10^4, 10^{11}]$ | 1.03 | $N^c$ & $N^c$ | N |
| Geant4L | $[10^2, 10^{12}]^b$ | 1.17 | $Y^d$ & Y | N |
| Geant4D | $[10^2, 10^{12}]^b$ | 1.00 | $Y^d$ & Y | N |
| GRRR | $[10^4, 10^7]$ | 12.4 | $Y^d$ & Y | Y |
| MCPEP | $[10, 10^8]$ | 102 | N & Y | N |

[a] 10 keV is the lowest energy advised in the manual, but in this study we found that this is too low, see Sect. 5.3.

[b] 250 eV minimum for electrons and positrons and 10 eV minimum for photons.

[c] Not out of the box, but there are add-ons or umbrella codes with provide this feature, for example CORSIKA (Heck et al., 1998).

[d] The magnitude of the electric field will be limited by the choice of the low energy cut-off.

Here we focus on the second problem, namely the beam propagation towards detectors where the final products are characterized by energy spectra and arrival times, and the source properties must be reconstructed from this data, e.g. in the work by Østgaard et al. (2008). Accurately modeling the transport from the source to the very remote detector is, together with some knowledge of the source, thus very important to deduce production altitude, beaming angle or light curves of TGFs and associated electron beams from space data (Dwyer and Smith, 2005; Carlson et al., 2007; Hazelton et al., 2009; Dwyer et al., 2008b; Sarria et al., 2016).

### 1.3 Content and order of the present study

To model particle beams in air far from the source, some researchers use general purpose Monte Carlo (MC) codes developed by large collaborations like Geant4 (used by Carlson et al. (2011) and by Skeltved et al. (2014)) or FLUKA (used by Dubinova et al. (2015)). On the other hand, to model, e.g., the radiation sources with their external or even self-consistent time dependent electric fields, other researchers develop custom made codes in small groups or as individuals, where the cross sections and numerical methods may come from already validated theory (e.g. Sarria et al. (2015); Kohn et al. (2014)).





While they are necessary for the understanding of the full physical phenomena, custom made codes are difficult to validate, especially if they are not made available by open access. Differences between one code and another may be explained by at least the following four factors:

- The choice of the included physics, as a compromise between correctness and feasibility.

- Cross sections, that can come from theory, measurements or both. In most cases the cross section data has a certain uncertainty.

- Numerical and coded implementation, e.g. numerical integrations, interpolations, roundoff errors and bugs.

- The performance, as faster codes can run more particles in the same time, which results in more accurate statistics.

Even if it is possible in principle to determine the differences between the physical models and between the numerical methods,
it may be very complicated (if not impossible)

- to estimate the uncertainties associated with a certain choice of physical models,

- to estimate the uncertainty propagation and accumulation of all input through the full multiscale models, and

- to review all source codes (if available) to find any mistakes and possible numerical problems.

In general it is found that software is underrepresented in high energy physics literature in spite of its significant contribution
to the advancement of the field (Basaglia et al., 2007).

Therefore, we here strive to provide a comparison standard for the particle codes, as simple and as informative as possible, by only considering their physical outputs. We have chosen standard tests for the core parts of all codes: the evolution of monoenergetic and monodirectional beams of photons, electrons and positrons through homogeneous air and without electric or magnetic fields. We elaborate our standard tests in the methodology section 4.

The targeted energy interval for high energy atmospheric physics in this study is from tens of keV to tens of MeV, bounded above by the observed maximal energy in a TGF (Briggs et al., 2010; Marisaldi et al., 2014). Typically a low energy cut-off is chosen for two reasons:

1. The codes developed for accelerator or cosmic-ray applications use typical energies well above 1 MeV, larger than the rest mass of electrons and positrons. For these energies relativistic approximations are accurate, ionization potentials are
negligible, and electron impact ionization is essentially a free-free elastic collision (i.e., similar to a collision of two free electrons). These approximations limit the validity of the codes at lower energies.

2. The mean free path of particles decreases and the number of particles increases with decreasing energy. Simulating with or without a low energy cut-off can make a difference of minutes to months of simulation time. Therefore a low energy cut-off is wanted for computational reasons.





The different implementations of the low energy cut-off, as reviewed in Sect. 3 cause significant differences in the results, see Sect. 5. These differences increase when electric fields are added, see Sect. 6 and puts an extra restriction on the value of low energy cut-off (Skeltved et al., 2014).

This paper is organized as follows: Sects. 2 and 3 review the particle interactions and the codes included in this study. Sect. 4 describes the methodology we used to compare the codes. Sect. 5 contains a discussion of important differences between the results of the tested codes, and in Sect. 6 the implications of adding electric fields are discussed. Finally we conclude and give a list of recommendations for High Energy Atmospheric Physics simulations in Sect. 7.

## 2 Overview of interactions and approximations

In High Energy Atmospheric Physics (HEAP) it is usually assumed that the density of the considered high energy particles is too low to directly interact which each other, therefore they only interact with the background medium which here are the air molecules. In addition for some 'self-consistent' codes, like GRRR (see Sect. 3.4), charged particles can interact non-locally due to the electric fields they produce. But for the present study these interactions are turned off, resulting in a linear problem. This means that the number of particles at the end of the beam is proportional to the particle number in the initial beam, and that different beams simply add up according to the superposition principle. Below we summarize the interactions considered for electrons, positrons and photons in HEAP. In these interactions the target molecule M and its resulting state are explicitly given, but for the MC model of the high energy particles, these molecules (or ions) act as a random background.

### 2.1 Electrons and positrons

Electrons and positrons above 50 keV (which is the low energy cut-off in our study) behave almost identically; they scatter elastically on molecules M, they ionize them, and they create bremsstrahlung on collisions with molecules:

$$e^{\pm} + M \rightarrow \begin{cases} e^{\pm} + M, & \text{elastic (Rutherford),} \\ e^{\pm} + e^{-} + M^{+}, & \text{ionization,} \\ e^{\pm} + \gamma + M, & \text{bremsstrahlung,} \end{cases} \tag{1}$$

with cross sections that only slightly dependent on the incoming particle type.

In addition, when positrons come to rest, they annihilate,

$$e^{+} + M \rightarrow 2\gamma + M^{+}, \quad \text{annihilation,} \tag{2}$$

and produce two photons of 511 keV. The standard implementation is that, when a positron drops below the low energy cut-off, it comes at rest immediately (in space and time). In reality the positron will come to rest over some distance and time, forming positronium (e.g. an $e^{+}e^{-}$ bound state), before annihilation. The positronium has a lifetime depending on the spins of the positron and electron (Karshenboim, 2004), forming a singlet or triplet state with lifetimes of 124 ps or 139 ns (in vacuum), respectively.





In the eV regime, the interactions are getting more complex, as molecular excitations and dissociations need to be taken into account explicitly.

### 2.1.1 Friction (or stopping-power) for electrons and positrons

Usually, the energy transfer in an ionization collision of electrons and positrons with molecules is of the order of 10 eV, hence it causes only a small energy loss for a particle with energy above the keV range. By introducing a so-called low energy cut-off $\varepsilon_{\text{cut}}$ , 'high' and 'low' energy particles and interactions can be decoupled. In this approximation, interactions producing secondary particles below the low energy cut-off are approximated as friction, while interactions with secondary particles above the cut-off are included explicitly.

Let $\varepsilon_1$ be the energy of the primary particle and $\varepsilon_2$ the energy of the secondary particle. The cross section $\sigma_k(\varepsilon_1)$ (in units of area) gives the probability of the primary particle to undergo an interaction labeled $k$. The differential cross section $\mathrm{d}\sigma_k(\varepsilon_1,\varepsilon_2)/\,\mathrm{d}\varepsilon_2$ (in units of area per energy) gives the probability of a primary particle to produce a secondary particle within the infinitesimal energy interval $[\epsilon_2, \epsilon_2 + d\epsilon_2]$ for the interaction $k$.

The secondary energy $\varepsilon_2$ can take values between the minimum $\varepsilon_{\text{min}}$ (of the order of eV and the primary is not sensitive for the precise value) and the maximum $\varepsilon_{\text{max}}$ (of the order $\varepsilon_1$), depending on the interaction. For ionization $\varepsilon_{\text{max}} = \varepsilon_1/2$ as the primary by convention is defined to be the final particle with the highest energy. For bremsstrahlung we have $\varepsilon_{\text{max}} = \varepsilon_1$.

Now the energy range of the secondary particles is decomposed into two parts: the first part from $\varepsilon_{\text{min}}$ to $\varepsilon_{\text{cut}}$ is implemented as a friction, and the second part from $\varepsilon_{\text{cut}}$ to $\varepsilon_{\text{max}}$ is implemented by discrete collisions.

The friction $F_k$ of interaction $k$ is defined as

$$F_k(\varepsilon_{\text{cut}},\varepsilon_1) = N \int\limits_{\varepsilon_{\text{min}}}^{\varepsilon_{\text{cut}}} \left( \varepsilon_{\text{loss}}(\varepsilon_2) \frac{\mathrm{d}\sigma_k(\varepsilon_1,\varepsilon_2)}{\mathrm{d}\varepsilon_2} \right)\ \mathrm{d}\varepsilon_2, \tag{3}$$

where $N$ is the number density of molecular collisions targets M, and $\varepsilon_{\text{loss}}$ the energy loss of the primary which is of the order of $\varepsilon_2$ plus the ionization energy. The resulting friction on the primary is given by the sum of all considered interactions,

$$F(\varepsilon_{\text{cut}},\varepsilon_1) = \sum_k F_k(\varepsilon_{\text{cut}},\varepsilon_1). \tag{4}$$

For electrons and positrons in the energy regime important for HEAP, the resulting friction is almost completely determined by the ionization part, as illustrated in Fig. 1. Especially if only the friction with $\varepsilon_{\text{cut}} = 50\,\text{keV}$ is considered (solid line), there the energy loss due to bremsstrahlung is more than two orders smaller than the energy loss due to ionization.

We remark that the friction is also frequently called the stopping-power for historical reasons, though it has the dimension of friction (energy/length) rather than of power (energy/time).

### 2.1.2 Straggling

In a simple implementation of the low energy cut-off, the primary particle suffers a uniform (and deterministic) friction $F(\varepsilon_{\text{cut}},\varepsilon_1)$, as given in Eq. (4). This means that now only the energy of the primary particle is altered, but not its direc-





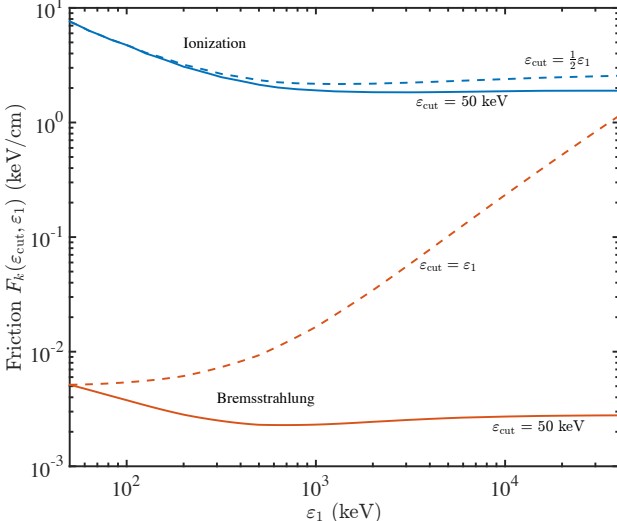

**Figure 1.** Friction $F_k(\varepsilon_{\mathrm{cut}}, \varepsilon_1)$ for electrons per interaction (Bremsstrahlung in red and ionization in blue), for two different low energy cut-offs, $\varepsilon_{\mathrm{cut}} = 50$ keV (solid line) and $\varepsilon_{\mathrm{cut}} = \varepsilon_{\mathrm{max}}$ (dashed line). The resulting friction is the sum of the two contributions, which in the energy regime of HEAP the resulting friction is dominated by the ionization (please, note the log-scale). The data are from (Cullen et al., 1997; Perkins et al., 1991) for an air density of $1.293 \times 10^{-3}$ g cm$^{-3}$ corresponding to 1 bar and 273 K as used in this study.

tion. A greater concern is that the accuracy of the assumed uniform energy loss is a matter of length scale. If the scale is much smaller than $\varepsilon_1/F(\varepsilon_{\mathrm{cut}}, \varepsilon_1)$, only a few interactions have taken place. On such a small length scale the real energy loss distribution (if one had considered all interactions explicitly) among the population would have a large spread. This effect is called straggling, and it was first studied by Bethe and Heitler (1934).

5   One way to mimic the real energy distribution is by implementing a stochastic friction, as is done in FLUKA and Geant4L. Basically the energy loss of the primary particle is as if it would be modeled by real low energy collisions below the cut-off, but without creating the secondary particles and without altering the direction of the momentum. The different implementation of the low energy cut-off (i.e., different implementations of the friction) is one of the significant differences we see in the studied programs, as discussed in the results section 5.

10   **2.1.3   Continuous Slowing Down Approximation**

Using the friction equation (3) over the whole range of secondary particle energies, hence with $\varepsilon_{\mathrm{cut}} = \varepsilon_{\mathrm{max}}$, the expectation value of the maximal penetration depth of a primary particle into a medium can be calculated in the so-called Continuous Slowing Down Approximation (CSDA). Integrating the friction over distance $\ell$ up to the point where the particle has lost all



its primary energy $\varepsilon_1$,

$$\int_{\ell(\varepsilon_1)}^{\ell(0)} F(\varepsilon_{\max}, \varepsilon(\ell)) \, \mathrm{d}\ell = \int_{\varepsilon_1}^{0} F_{\mathrm{tot}}(\varepsilon_{\max}, \varepsilon) \, \frac{\mathrm{d}\ell}{\mathrm{d}\varepsilon} \, \mathrm{d}\varepsilon = \varepsilon_1, \tag{5}$$

defines one CSDA range through

$$\mathrm{CSDA}(\varepsilon_1) = \ell(\varepsilon_1) - \ell(0). \tag{6}$$

One CSDA range is thus the maximal length that primaries can penetrate into a material. Due to feedback from secondaries (e.g. electron -> photon -> electron) the complete avalanche can survive longer. As we describe in the methodology section 4, we choose half a CSDA range as the optimal detector distance to compare the differences in outputs of the codes as comprehensively as possible.

## 2.2 Photon interactions

The typical photon interactions are

$$\gamma + \mathrm{M} \rightarrow \begin{cases} \gamma + \mathrm{M}, & \text{elastic (Rayleigh),} \\ e^- + \mathrm{M}^+, & \text{ionization (by absorption),} \\ \gamma + e^- + \mathrm{M}^+, & \text{ionization (by Compton),} \\ e^+ + e^- + \mathrm{M}, & \text{pair production.} \end{cases} \tag{7}$$

Photons have no charge, and therefore they lose energy much less gradually than electrons and positrons. In a typical inelastic interaction of a photon, the energy loss is significant.

### 2.2.1 Photon attenuation

The most important interaction for low energies (below 30 keV) is photo-absorption, and for the highest energies (above 40 MeV) it is pair production; in both cases the photon completely disappears. Inbetween, where Compton scattering is most important, the energy loss per interaction is still significant; the expectation value for the energy loss of the primary photon grows from 5% (at 30 keV) to above 90% (at 1 MeV). The Continuous Slowing Down Approximation is thus not appropriate for photons, as photons do not continously lose small amounts of energy, in contrast to electrons and positrons, but they lose a substantial fraction of their energy after some free path. Consecutively, for most energies (certainly above 1 MeV and below 30 keV) the photon intensity $I$ can be approximated by an exponential decay or attenuation,

$$I(\ell) = I(0) \exp(-\ell/\mu), \tag{8}$$

where $\mu(\varepsilon)$ is the attenuation-coefficient depending on energy (and material).

In this work we need to estimate an appropriate detector distance (the exponential decay does not appear explicitly in any model), and we use two e-folding lengths (i.e., the inverse of half the attenuation-coefficient) as the optimal detector distance to compare the output differences, as described further in the methodology section 4.





## 3 Overview of codes

In Table 1 we have summarized the codes used in this benchmark. In this chapter we give more detailed descriptions.

### 3.1 EGS5

EGS5 (Electron-Gamma Shower version 5, developed by Hirayama et al. (2005)) is a general purpose software package for

the Monte Carlo simulation of the coupled transport of electrons, positrons and photons in an arbitrary geometry. It is the next version after EGS4 that was released by Nelson et al. (1985) with a history that dates back to 1960's. The user controls an EGS5 simulation by means of an input text file for settings and a written FORTRAN user code, to which the rest of the FORTRAN source files are appended and compiled as one. In the user code several subroutine calls create, establish and initiate the cascade. Two important subroutines HOWFAR & AUSBGAB, which should be written inside the user-code are to

specify the geometry and the output of the results. EGS5 can simulate particles from a few keV up to several hundred GeV, depending on the material. There is a limited option for including magnetic fields, and no option to include electric fields. All interactions of equations (1), (2), and (7) are implemented, in this work with a low energy cut-off of 50 keV. In the user manual of Hirayama et al. (2005) a minimum low energy cut-off of 10 keV is advised, but we noticed that for the bremsstrahlung cross sections relativistic limits are applied, which results in a too low production of photons, see Sect. 5.3. Friction is implemented

uniformly, without straggling effect (that is to say without fluctuations in the energy loss). The input file and user code, used in this work, can be found in the supplementary material. Please see the documentation of Hirayama et al. (2005) for a detailed overview of the implemented physics.

### 3.2 FLUKA

FLUKA (developed by Ferrari et al. (2005), copyright to INFN and CERN 1989-2011), is a general purpose tool for calcula-

tions of particle transport and interactions with matter. FLUKA is able to simulate the interaction and propagation in matter of roughly 60 different particles, including photons from 100 eV and electrons and positrons from 1 keV to thousands of TeV, neutrinos, muons of any energy, hadrons of energies up to 20 TeV (up to 10 PeV by linking FLUKA with the DPMJET code) and all the corresponding antiparticles, and neutrons down to thermal energies. FLUKA includes recent datasets, published by Böhlen et al. (2014). The program can handle magnetic and electric fields, although not self-consistently (i.e., the charged

particles do not produce magnetic or electric fields). The program, written in FORTRAN, reads in so called user-cards, in which the user defines the geometry, materials and detectors. The user card, used in this work, can be found in the supplementary material. All interactions of equations (1), (2), and (7) are implemented, f in this work with a low energy cut-off of 50 keV. Friction in FLUKA is modeled with universal fluctuations, mimicking the straggling effect, meaning that the primary particle loses its energy as if it would undergo random collisions. But the direction of its momentum is not changed and no

secondary particles are produced. Please see the documentation 'FLUKA Manual' at www.fluka.org for a detailed overview of the implemented physics.





### 3.3 Geant4

Geant4 is an open source toolkit to simulate the passage of particles through matter, developed by a wide international collaboration lead by the CERN. It is coded in C++, following an object oriented philosophy. It can simulate the transport of almost all known particles, and can include electric and magnetic fields (Agostinelli et al., 2003). We use the version 10.2 released in December 2015. In Geant4, the user can choose between six main models for the treatment of electrons, positrons and photons, with different performances and accuracies. One can also specify the implementation of the friction, to take into account energy losses below the low energy cut-off. For this study we are using two Geant4 configurations, that are detailed below. All Geant4 codes are available in the supplementary material. References and details for these models are presented in the 'Geant4 Physics reference manual' available at http://geant4.web.cern.ch.

#### 3.3.1 Geant4D

Geant4D uses the default model, but in addition we deactivated the fluctuations of the continuous energy loss, i.e. the energy losses are applied uniformly, without straggling effect. This choice is for benchmark purposes, to identify the effect of straggling.

#### 3.3.2 Geant4L

Geant4L uses the Livermore model, which uses cross sections from the EPDL and EEDL databases, provided by the Lawrence Livermore National Laboratory. The detailed implementation is provided in (Cullen et al., 1997; Perkins et al., 1991). The 'Universal fluctuation model' is activated to include the straggling effect in the implementation of friction.

### 3.4 The GRanada Relativistic Runaway (GRRR) code

Developed by A. Luque at the Astrophysics Institute of Andalusia (IAA-CSIC), the GRanada Relativistic Runaway (GRRR) code was designed to investigate the self-consistent interaction between electrons in the limit of very intense Relativistic Runaway Electron Avalanches (RREA). This investigation, presented in Luque (2014), concluded that due to the interaction between electrons in the avalanche RREAs saturate into a steady-state propagating Relativistic Runaway Ionization Front (RRIF). As the GRRR code was implemented with that specific goal in mind, its scope is narrower than the general purpose codes (EGS5, FLUKA, Geant4) analyzed in this paper. It only follows the evolution of high-energy electrons, and includes a limited set of interactions between these electrons and the embedding medium. Electron ionization and Rutherford scattering are modeled discretely, and in this work down to a low energy cut-off of 50 keV. The friction for these interactions is uniform, without straggling effect. Bremsstrahlung collisions with nuclei are modeled deterministically by friction, in other words: as continuous radiative losses. The supplemental material of Luque (2014) contains further details about the physical model underlying the GRRR code. In the supplement material of this work the input files are given for the presented benchmark tests. The full source code for GRRR is available at https://github.com/aluque/grrr. However, presently the code is mostly undocumented so we advise potential users to contact the author.



## 3.5 MC-PEPTITA

The Monte Carlo model for Photon, Electron and Positron Tracking In Terrestrial Atmosphere (MC-PEPTITA) by Sarria et al. (2015) is a Fortran 90 code that simulates the propagations of TGF and associated electron/positron beams within the Earth environment, from the production altitude at 10 to 20 km to satellite altitude. To simulate the quasi-exponential atmospheric

density profile and the Earth's magnetic field, it uses the NRLMSISE-00 and IGRF-11 models (Cullen et al., 1997; Perkins et al., 1991). It is optimized to run in this environment, whereas some other codes (e.g., Geant4) can only handle layers of constant density. Concerning the interactions between particles and matter, it mainly uses the EPDL and EEDL cross section sets (Cullen et al., 1997; Perkins et al., 1991), except for inelastic scattering of electrons and positrons where the GOS model is used. The interactions are simulated similarly to PENELOPE (Salvat et al., 2011), with equivalent numerical methods. MC-

PEPTITA does not include any continuous energy losses: the particles are followed discretely down to the lowest possible energies allowed by the models used, with exception of bremsstrahlung where the minimal energy is set to 100 eV.

## 4 Methodology

We focus on the evolution of monoenergetic and directed beams of electrons, positrons and photons with kinetic energies between 100 keV and 40 MeV through homogeneous air in the absence of electric and magnetic fields, using a low energy

cut-off of 50 keV. Providing a first benchmark, in the case when the fields are turned off. Assuming sufficiently low densities of high energy particles, arbitrary particle beams can be decomposed into such monoenergetic and directed beams.

The electron, positron and photon beams propagate through air, consisting of 78.085% nitrogen, 20.95% oxygen and 0.965% argon. We use a constant and homogenous air density of $1.293 \times 10^{-3}$ g cm$^{-3}$ which is corresponds to 1 bar and 0 degree Celsius. For all programs we choose a low energy cut-off of 50 keV, below which all particles are removed. For most programs,

this low energy cut-off is also the threshold to treat collisions discretely or continuously, with two exceptions: MC-PEPTITA handles all collisions explicitly, and GRRR uses continuous radiative loss (bremsstrahlung). During the simulation electrons, positrons or photons above the low energy cut-off can be created (except for GRRR, which only models electrons), and are then followed as well until they also drop below the low energy cut-off. If considered in the program, positrons dropping below the low energy cut-off can produce photons by annihilation above the low energy cut-off.

We use ideal flat surface detectors, perpendicular to the primary particle beam. On a detector, the type, kinetic energy, position and arrival time of the arriving particles are recorded. After detection, the particles are removed from the program, thus we do not measure backscattered particles that have already been detected. Depending on the program, other secondary particles are created with a very low probability (e.g. neutrons by photo-nuclear interactions), but we do not record them in the output. First, we study the particle number of all particles as function of propagation distance (attenuation). Second, for

one specific distance, (depending on particle type and initial energy) we proceed to a detailed analysis of energetic, spatial and temporal distribution. Complementarily we also benchmark the performance (i.e., the simulation completion time) of the programs used in this study.



### 4.1 The number of particles versus distance (attenuation)

We study the particle number of all particles as a function of beam propagation distance, up to of one CSDA range for electrons and positrons and of four times the inverse of the attenuation coefficient (four e-folding lengths) for photons. This range is divided in several distances (roughly 20) or data points. For each distance (or data point), we perform a new simulation. Each

simulation with ten thousand particles in the initial beam, for beams of electrons, positrons and photons with energies of 0.1, 0.4, 1, 10 and 40 MeV. The particle numbers are therefore derived under the assumption that the detectors are impenetrable. This means that back scattering is excluded, and that the particle number therefore is lower than in a passing avalanche in air only.

We added a $\pm 1/\sqrt{n_i}$ relative error expected from the Monte Carlo methods ($n_i$ being the number of counts in the $i$th bin).

In this way we performed roughly 1800 simulations, namely circa 300 simulations per program: for 3 particle types, 5 initial energies and on average 20 distances per beam. GRRR only considers electrons while the energy loss due to production of photons is implemented as a continuous energy loss. The relevant results are given and discussed in Sect. 5. In addition all data of this part are visualized and available in the supplementary material.

### 4.2 Spectral analysis

We performed detailed simulations with 1 million particles per beam for one specific distance per beam. For electrons and positrons, the detection distance was chosen as half of the CSDA range. This gives most information in one plot, since the primary particles are still alive, while there is a significant number of secondary particles produced. For photons, the inverse of half the attenuation coefficient (two e-folding lengths) is chosen as the distance for the detailed study. At the detector we analyze the kinetic energy, the radial distance from the symmetry axis and the time of arrival. The spectra are binned using the

Freedman–Diaconis rule in the log-domain and rescaled to numbers per primary. As also for the attenuation study, we added a $\pm 1/\sqrt{n_i}$ relative error expected from the Monte Carlo methods ($n_i$ being the number of counts in the $i$th bin). We performed roughly 90 different simulations (circa 15 simulations per program: 3 particles and 5 initial energies). The relevant results are given and discussed in Sect. 5. In addition all data of this part are visualized and available in the supplementary material.

### 4.3 Performance benchmark

As a complement, we also tested how much time the codes needed to complete the simulations. We did not try to do an in-depth performance benchmark of the codes, but we think this is an interesting piece of information for someone who is seeking for a code to be used in the HEAP context. Since the programs are written in different languages (Fortran, C++ and Python) and may be run on different machines with different architectures, we normalized the completion time with respect to a reference computer configuration.

The simulation starting with one million 1 MeV electrons is used as the test case because it is feasible for all the evaluated codes, and it takes a completion time that is neither too short, nor too long. More details are given in the supplementary material. The normalized results are discussed in Sect. 5.5.



## 5 Results

Most tests show similar outputs for the different codes, to within deviations of $\pm 10\%$, see supplementary material. Here we focus on important differences between the results of the codes, and we provide several plausible explanations.

### 5.1 Straggling

For electrons and positrons below 1 MeV, the data clearly show the effect of straggling, as discussed in Sect. 2.1.1). For example in the 400 keV electrons beam shown in Fig. 2, EGS5, Geant4D and GRRR do not include straggling, therefore the maximal electron energy is too small and the drop of the energy spectrum towards this maximal energy too steep. Geant4L, MCPEP and FLUKA show the correct spectrum, but for different reasons. MCPEP simulates without a low energy cut-off (and thus without friction). Geant4L and FLUKA use a stochastic implementation of the friction called universal fluctuations.

Basically the friction is not applied uniformly to all particles of the same energy equally, but a distribution of energy losses in time mimics the random nature of the collisions. Only the direction change is considered negligible.

The same effect is also seen for electron and positron beams with energy above 10 MeV, in the scenario where bremsstrahlung is treated as continuous. GRRR shows an unphysical drop in the electron spectrum at high energies, as illustrated in Fig. 3. The reason is that the energy loss by bremsstrahlung is mostly above the low energy cut-off, see Fig. 1, meaning that the energy

loss of the electrons and positrons is mostly due to discrete 'hard' collisions and thus ill-approximated by uniform averaged friction. Nevertheless we found that the total integrated energy is similar. This approximation is also used by others in the community like Celestin et al. (2012); Chanrion et al. (2014).

### 5.2 Opening angle

High energy photons penetrate the medium much deeper than electrons and positrons, and therefore small differences in

opening angles after Compton collisions are more important. In inelastic collisions photons always lose a significant amount of energy, as discussed in Sect. 2.2, and therefore they get a significant opening angle.

MCPEP simulates all collisions explicitly (others use a friction - which does dot change the primary direction). The energy spectra agree between these codes, but Fig. 4 illustrates, that the radial and temporal spectra vary: MCPEP shows a wider photon beam and substantially later photon arrival times.

### 5.3 Bremsstrahlung


We saw that EGS5 uses an ultra-relativistic approximation in the treatment of bremsstrahlung and thereby we question the validity at lower energies, as discussed in Sect. 3.1). For the primary electron, in the energy regime important for HEAP, bremsstrahlung is negligible compared to ionization (see Fig. 1) and we thus do not see a difference there, but in the production of photons there is a significant difference, as can be seen in Fig. 5.





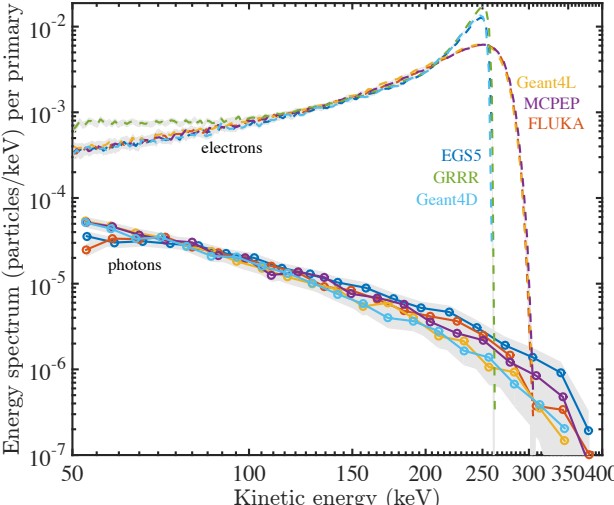

**Figure 2.** Products of a beam of 400 keV electrons after a propagation distance of 0.5 times their CSDA range which is 1.9 m in air at 1 bar and 273 K. The electrons have now a maximal energy of 250 to 300 keV depending on the code, but the total integrated energy is equivalent. The difference in electron distribution is due to straggling by ionization, see Sect. 5.1

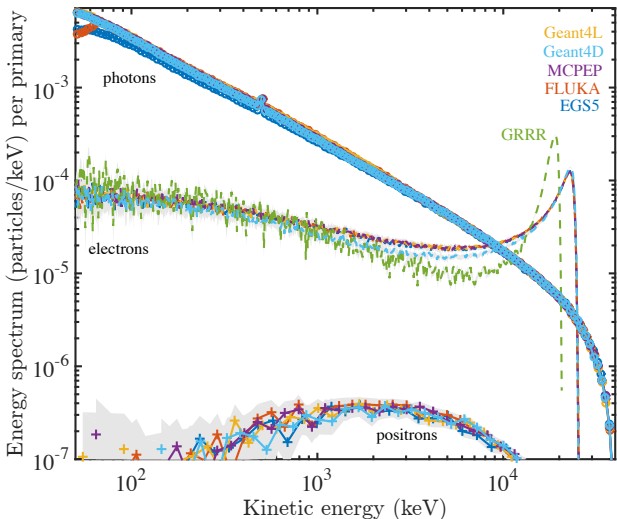

**Figure 3.** The same as in Fig. 2, but now for 40 MeV electrons. The propagation distance of 0.5 times their CSDA range is now 63.8 m (1 bar and 273 K). Now not only electrons and photons, but also positrons have been produced. The difference in electron distribution is due to straggling by bremsstrahlung, see Sect. 5.1



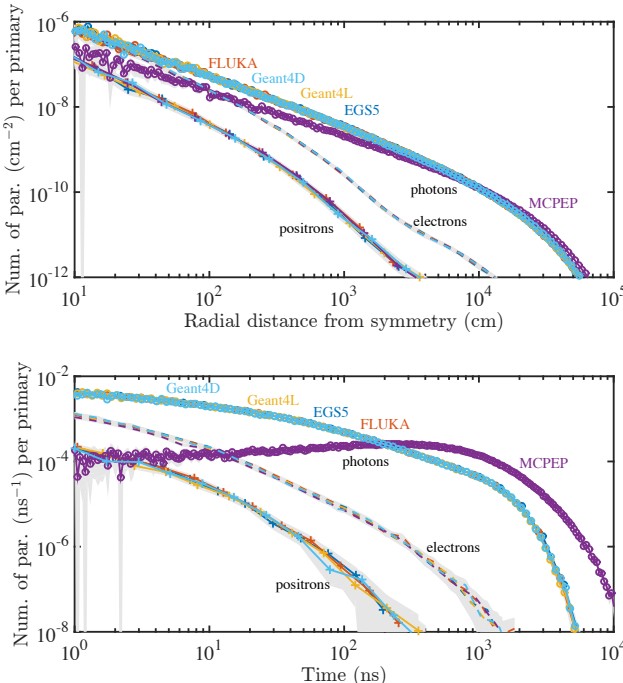

**Figure 4.** Products of a beam of 10 MeV photons at a distance of $1/(0.5\ \mu)$ which corresponds to 756 m (1 bar and 273 K). Particle number per primary as a function of the radial distance from the symmetry axis (above), and of arrival time (below).

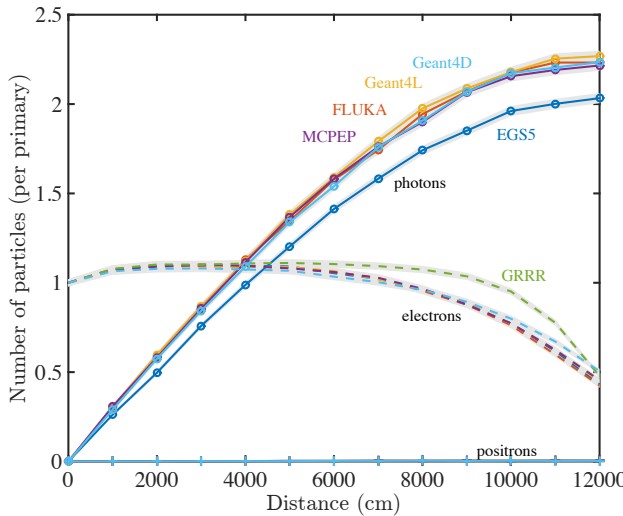

**Figure 5.** Products of a beam of 40 MeV electrons, as detected by 12 detectors at 10 to 120 m distance in 1 bar and 273 K. The detectors are impenetrable to hinder backscattering; therefore a new simulation is run for every detector distance.





## 5.4 Other differences

Other differences we have found are listed below.

– For the electron and positron beams we see in the energy spectrum of FLUKA below 70 keV a dip in the number of photons. Fig. 3 shows an example.

– For the electron beams $\leq 1$ MeV (but not in the positrons or photon beams) we see a difference in the longest arrival times ($> 100$ ns) for photons between the programs FLUKA and EGS5 compared to Geant4D and Geant4L. GRRR does not model photons, and MCPEP is completely different because of the opening angle, see Sect. 5.2.

– GRRR shows a slight higher count (less than $15\%$ higher) than the other codes for the number of electrons in the avalanche as function of distance. Fig. 5 shows an example. In the energy spectrum we see that these electrons are in the
low energy tail of the spectrum, see for example Fig. 2.

– For the electron and positron beams we see a difference in the shortest arrival times ($< 1$ ns) for electrons and positrons between the programs FLUKA,EGS5 and MCPEP compared to Geant4D, Geant4L and GRRR.

## 5.5 Performance

The performances in terms of completion time of the codes are presented in Tab 1. On one hand, we see a clear difference
of performance between MC-PEPTITA (simulations with a low energy cut-off as low as possible) and the rest. As said in the introduction, the low energy cut-off is generally introduced to speed up the simulation. Moreover, MC-PEPTITA was not optimized to run with a constant density and without magnetic field, and is then making a significant amount of useless (but time consuming) calculations for this benchmark case. On the other hand, the choice to simulate all particles synchronously (to include self-consistent electric fields) slows the simulations significantly down, as seen for GRRR.
Concerning codes developed by wider collaborations, Geant4 and FLUKA show similar and best performances, but EGS5 is about 4 times slower. We can also note that in Geant4, the use of the energy straggling costs about 20% more computation time than turning it off.

## 6  The effect of electric fields

In this study we have provided benchmarks in the absence of electric fields, applicable to custom codes when the fields are
turned off. The programs reviewed in this study are at least able to simulate the simplest case of particle beam evolution in air, in the wide distance from the particle source to detectors in space and on ground. However, as discussed in the introduction, the particles are initially accelerated by electric fields in the thunderstorm, either by weaker fields in larger regions in the Relativistic Feedback regime, or by strong and very localized self-consistent electric fields in the Cold Runaway regime. We here give a short outlook on the range of validity of the presented models in these cases. In general, it can be expected that
electric fields will magnify all differences in choice and implementation of cross-sections to a certain extent, because particles



not just lose energy and drop eventually below the energy cut-off, but charged particles can also be reaccelerated and reappear in the ensemble.

To be specific, we recall the definition of the three characteristic electric fields and electron energy regimes of the problem (giving field values for air at standard temperature and pressure (STP)). For electrons with energies in the eV regime, the

classical breakdown field is $E_k \approx 32$ kV/cm. For higher fields electron avalanches are formed, but their energies typically do not exceed the range of several eV, as their friction increases with energy. The electron friction increases up to an electron energy of approximately 200 eV where the critical electric field $E_c \approx 260$ kV/cm is required to balance friction — as long as the approximation of the electron ensemble by classical friction is valid. For electron energies above 200 eV the friction decreases to a minimum that is balanced by an electric field of $E_b \approx 2.8$ kV/cm, called the break-even field, at an electron

energy of about 1 MeV.

Clearly two limitations to using a particle model with a low energy cut-off are immediately visible. First, if the electric field is above the critical electric field of 260 kV/cm ($E > E_c$) in a sufficiently large volume, the two populations of electrons with energies below and above 200 eV are strongly coupled and essentially all electrons can be accelerated into the runaway regime, to 1 MeV and beyond. Second, if the electric field is below the critical field, but above the classical breakdown field

($E_k < E < E_c$), the population of electrons in the eV regime (the so-called thermal electrons) can grow strongly, and eventually 'tunnel' into the run-away regime; we will come back to this effect below.

On the other hand, for electric field strengths below the break-even field ($E < E_b$), all electrons, regardless of initial energy, will eventually stop as the friction force of air is stronger than the accelerating force of the electric field.

Finally, when the electric field is above the break-even and below the classical breakdown field ($E_b < E < E_c$), the use

of the energy cut-off of 50 keV (or even lower) can have strong implications: For an electron energy of 50 keV, friction and electric acceleration force balance each other when the field is 7.8 kV/cm. So in classical approximation one would estimate that at lower fields the inclusion of the cut-off is justified. However, this classical approximation neglects the stochastics of the actual process. Due to the randomness of free paths and scattering events, electrons actually can 'tunnel' into energy regimes that they could not reach in the classical approximation, an effect similar to the straggling effect discussed earlier.

Skeltved et al. (2014) have observed this effect: For all fields between 4 and 25 kV/cm, they found that energy spectrum and mean energy of runaway electrons depended on the low energy cut-off, even when it was chosen between 250 eV and 1 keV. They also found – not surprisingly – that the differences become most apparent when the electric field force approaches the friction force corresponding to the low energy cut-off.

A related observation was made by Li et al. (2009) when they found electron runaway from a negative streamer even though

the maximal electric field at the leader tip was well below the critical field $E_c$.

Future studies on how to choose the low energy cut-off for given fields are desirable to optimize computations between efficiency and accuracy.



## 7 Conclusions

The goal of this work is to provide standard tests for comparing the core part of Monte Carlo simulations tools available for HEAP. We focused on the propagation of electrons, positrons and photons through air, in the absence of electric and magnetic fields. We compare the output at half the CSDA range for electrons and positrons, and at two e-folding lengths (the inverse of half the attenuation coefficient) for photons. We have run these tests for 0.1, 0.4, 1.0, 10 and 40 MeV initial energy for the several codes (Geant4, EGS5, FLUKA, GRRR, and MC-PEPTITA) used by the co-authors. The outputs show equivalent results, but there are important differences one can identify. Especially the different implementations of the friction are causing observable effects. First we see that straggling is important in the energy regime of HEAP and should be included in the simulations. Secondly the opening-angle of photon beams are very sensitive to the low energy cut-off. Thirdly we noticed that EGS5 has an ultra-relativistic approximation for bremsstrahlung which is not appropriate in the energy regime of HEAP. Last but not least there is a big difference in completion time between programs, mainly depending on the low energy cut-off and the synchronous implementation of the code. Adding electric fields will only increase these differences further and limits the value of the low energy cut-off. All results are published as supplementary material, and they can then be used by anyone to benchmark their custom made codes, with the fields switched off. The next step is provide benchmarks including fields and finding the optimal low energy cut-off for simulations in HEAP.

## 8 Recommendations

- Check where possible custom made codes to well established general purpose codes, we provide benchmarks in the energy regime of HEAP, in the case of zero field.

- Make your custom made code available to other researchers.

- For electrons and positrons below 1 MeV straggling should be included.

- For electrons and positrons above 10 MeV radiative loss should not be implemented with uniform friction.

- Photon production (due to bremsstrahlung) by electrons and positrons in energy regime of HEAP is under-estimated by EGS5.

## 9 Code and/or data availability

Figures of all output are available in the supplementary material. All raw data, circa 2 gb in compressed form, can be downloaded on request. In addition, the input files for reproducing the tests done in this benchmark are given for EGS5, FLUKA, Geant and GRRR, including links to the main source files. MC-PEPTITA simulations can be requested, contact David Sarria (david.sarria.89@gmail.com). MC-PEPTITA program was developed under a contract of Centre National D'Edtudes Spatiales (CNES) and Direction Générale de l'Armement (DGA), whose permissions are required in order to get access to the source



code. Details of the performance tests are also available in the supplementary material, including the reference code 'pidec.cpp', used for normalizing the different computer architectures.

*Author contributions.* CR and DS designed the tests with contributions of AS, AL and GD. CR, DS, AL and GD carried them out and discussed the differences. CR, DS, AS and UE prepared the manuscript with contributions from all co-authors.

5  *Acknowledgements.* Working visits between the European partners were supported by the ESF research training network TEA-IS (Thunderstorm Effects on the Atmosphere-Ionosphere System). Rutjes is funded by the Foundation for Fundamental Research on Matter (FOM), which is part of the Netherlands Organisation for Scientific Research (NWO). Skeltved is supported by the European Research Council under the European Union's Seventh Framework Programme (FP7/2007-2013)/ERC grant agreement 320839 and the Research Council of Norway under contracts 208028/F50, 216872/F50, and 223252/F50 (CoE). Luque was supported by the European Research Council (ERC) under
10  the European Union's H2020 programme/ERC grant agreement 681257 and by the Spanish Ministry of Economy and Competitiveness, MINECO under projects ESP2013-48032-C5-5-R and FIS2014-61774-EXP that include EU funding through the FEDER program. Diniz is financial supported by the Brazilian agencies CAPES and CNPq.



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
