# Peer review of "Evaluation of Monte Carlo tools for high energy atmospheric physics"

_Geoscientific Model Development, 2016_

## Referee Comment (RC1) · A Chilingarian (Referee) · 4 Jul 2016

**Referee comments on the paper submitted to Geoscientific Model Development (GMD) "Evaluation of Monte Carlo tools for high energy atmospheric physics" by Casper Rutjes, David Sarria, Alexander B. Skeltved, Alejandro Luque, Gabriel Diniz, Nikolai Østgaard, and Ute Ebert**

*A.Chilingarian, Yerevan Physics Institute, Armenia*

The paper concern problem of code verification, i.e., it answers to the questions is a code error-free, well engineered, meet specifications, are the test result compatible with analogical codes, and so on. Verification will not ensure that the system is useful for understanding the physical phenomenon under investigation and will not be crucial to choose the theory that explains experimental measurements in the best way. The procedures trying to answer the question will be the implementation of a code useful for understanding physics has an overall name of the code validation. Validation is the process of determining the degree to which a model, simulation, or federation of models and simulations, and their associated data are accurate representations of the real world (Dept. of Defence doc., 2008).
By model we mean a complete probability statement of what currently supposed to be known a priori about the mode of generation of data and of uncertainty about the parameters (Box, 1984). These definitions emphasized that simulations should confront the experimental data! The model and simulations should be architecture in a way to allow direct comparisons of experimental data and model results. Sure, the attention should be made on proper selection and quality control of experimental data as well.

The most difficult and most important part of the model validation is the comparison of competitive hypotheses and decision making on the nature of the investigated physical phenomenon. The new emerging field of high-energy atmospheric physics (HEAP) includes 2 main physical phenomena: Terrestrial Gamma Flashes (TGFs) - brief burst of gamma radiation (sometimes also electrons and positrons) registered by the orbiting gamma ray observatories in the space and Thunderstorm ground enhancements (TGEs) - the prolonged particle fluxes registered on the ground level|. There are an alternative name "Gamma glows" introduced by Joe Dwyer, but this name does not fit very well because on the ground we detect as well fluxes of electrons and neutrons. Both TGFs and TGEs are related to the thunderstorms and lightnings: TGEs - by directly detecting electric field and lightning occurrences above the detector site; TGFs by making rather complicated synchronization with worldwide lightning detecting networks.

The central engine initiated TGF and TGE is believed to be the Relativistic Runaway Electron avalanches (RREA) accelerated seed electrons in the terrestrial atmosphere up to 30-40 MeV. The *in situ* observation of numerous RREAs during strong thunderstorms on Aragats and first simultaneous measurements of TGE electrons and gamma ray energy spectra proved that RREA is a robust and realistic mechanism for electron acceleration. Detailed measurements of the gamma ray energy spectra by large NaI spectrometers on Aragats allow to reliably extending energy range of the "thunderstorm" gamma rays up to 100 MeV due to another "thunderstorm" gamma ray production mechanism - MOdification of the electron energy Spectrum (MOS).

Thus, the RREA mechanism operating in the lower and upper atmosphere generates 2 phenomena – the fluxes of electrons, gamma rays and neutrons on the Earth's surface, i.e. TGEs; and gamma rays and sometimes also electron/positrons in the space observed by the orbiting gamma ray observatories from the Earth's direction (TGFs).

TGFs and TGEs share many common features, as they are results of RREA. The drastic time difference (minutes for TGE and hundred of microseconds for TGF) is not essential because prolonged TGEs are nothing more than a superposition of the short nanosecond scale avalanches, which Aragats group has named Extensive cloud showers (ECS) and Alex Gurevich et. al., Micro runaway breakdown – MRB).

**2. Available Validation data from detected TGEs on Aragats**

The "natural electron accelerator" is operating in thunderclouds above the research station Aragats of Yerevan Physics Institute (Chilingarian, Hovsepyan and Mnatsakanyan, 2016) at an altitude of 3200 m on the plateau near a large lake. Numerous particle detectors and field meters are located in three experimental halls as well as outdoors; the facilities are operated all year round. All the relevant information is being gathered, including data on particle fluxes, fields, lightning occurrences, and meteorological conditions and is available via the multivariate visualization soft- ware ADEI on the Web page of the Cosmic Ray Division (CRD) of the Yerevan Physics Institute http://adei.crd.yerphi.am/adei. Several published papers provide information that can be directly compared with simulations. We present numerous energy spectra, intensities of gamma ray and electron fluxes, disturbances of electric field during TGEs; relations of lightnings and particle fluxes; relations of neutron and gamma ray fluxes and many others. Numerical data and plots on following research topics can be downloaded from the site:

- Cold Runaway or/and electron acceleration in the electric fields of the thundercloud.  TGE research do not support the first hypothesis; during large TGEs lightning activity is suppressed, lightnings stop particle flux not initiate it (Chilingarian et al, 2011, Hovsepyan and Kozliner , 2015);
- *In situ* measurements of RREA, density and space distribution of avalanches, etc (Chilingarian et al., 2011);
- Estimated phenomenological parameters of the RREA (Chilingaria, Mailyan and Vanyan, 2012);
- TGEs and Charge structure of Thundercloud (Chilingarian & Mkrtchyan, 2012, Chilingarian, 2014);
- Energy spectra of the TGE gamma rays (Chilingarian et all., 2013, Chilingarian, Hovsyapyan and Kozliner, 2016).
- Relation of TGEs and Lightnings, Chilingarian et all, 2015, Chilingarian, Chilingaryan and Reymers, 2015).

**3. Conclusion**

In Introduction section should be corrected the definition of HEAP including references to TGE phenomenon.

Add recommendations on the code validation.

**Reference**

Box G. E. P., The importance of practice in the development of statistics, Techno-metrics, Volume 26, Issue 1, 1984.
Chilingarian, A., A. Daryan, K. Arakelyan, A. Hovhannisyan, B. Mailyan, L. Melkumyan, G. Hovsepyan, S. Chilingaryan, A. Reymers, and L. Vanyan (2010), Ground-based observations of thunderstorm-correlated fluxes of high-energy electrons, gamma rays, and neutrons, Phys. Rev. D, 82, 043009.

Chilingarian, A., Hovsepyan, G., Hovhannisyan, A, 2011. Particle bursts from thunderclouds: natural particle accelerators above our heads. Phys. Rev. D: Part. Fields 83 (6), 062001.

Chilingarian, A. and Mkrtchyan, H., Role of the Lower Positive Charge Region (LPCR) in initiation of the Thunderstorm Ground Enhancements (TGEs), Physical Review D 86, 072003 (2012).

Chilingarian, B., Mailyan, Vanyan, L., 2012. Recovering of the energy spectra of electrons and gamma rays coming from the thunderclouds. Atmos. Res. 114–115, 1–16.

Chilingarian A., Hovsepyan G., Kozliner L., 2013, Thunderstorm ground enhancements gamma ray differential energy spectra. Phys. Rev. D: Part. Fields 88, 073001.

A.Chilingarian, Thunderstorm Ground Enhancements - model and relation to lightning flashes, Journal of Atmospheric and Solar-Terrestrial Physics 107 (2014) 68–76.

Chilingarian, A., S. Chilingaryan, and A. Reymers (2015), Atmospheric discharges and particle fluxes, J. Geophys. Res. Space Physics, 120, 5845–5853,.

Chilingarian A., Hovsepyan G., and Mantasakanyan E., 2016. Mount Aragats as a stable electron accelerator for atmospheric High-energy physics research, Phys. Rev. D: Part. Fields, 93, 052006.

Chilingarian A., Hovsepyan G., Kozliner L, Extensive Air Showers, Lightning, and Thunderstorm Ground Enhancements, Astroparticle Physics, 82 (2016) 21–35.

Dept. of Defence doc., 2008, Department of Defense Documentation of Verification, Validation & Accreditation (VV&A) for Models and Simulations". Missile Defense Agency. 2008

---

## Author Comment (AC1) · 8 Jul 2016

Reply to the referee report of Dr. A. Chilingarian (reference: gmd-2016-147-RC1)

We would like to thank the referee for his extensive report that concerns three subjects: 1. code verification and validation in the first paragraph of his report, 2. the phenomenon of Thunderstorm Ground Enhancements (TGEs) and their relation to the underlying physical mechanisms in the 2nd to 5th paragraph, and 3. available validation data from TGEs observed in Armenia and additional literature in the remaining comment.

Ad 1. We agree with the definitions of code verification and validation, but we would like to clarify, that we do not intend to validate any code in this paper, as the general purpose codes Geant4, EGS5 and FLUKA have already undergone multiple verification

and validation studies in different physical contexts, but they cannot readily be applied to all HEAP phenomena. Therefore, we rather provide here a set of tests – as simple and as informative as possible – to benchmark custom made codes that are built within the HEAP community.

Ad 2. The referee (Dr. A .Chilingarian) draws attention to Thunderstorm Ground Enhancements (TGEs), that have been measured by his group in Armenia. We agree that in our discussion of HEAP phenomena in section 1.1 we have used the terminology introduced by American and European researchers and did not mention the Armenian observations. The Armenian TGEs are similar (or the same?) as X-ray glows (high-energy photon emissions from a thundercloud with a time scale longer than 1 second) observed from planes (e.g. in the ADELE experiment), from balloons at thunderstorm altitude, or from ground at Mount Monju in Japan in winter time. TGEs can be short (<50 ms) or long events (> 1 second), but x-ray glows (as presented by Dwyer) are long events. Furthermore, long TGE events are supposed to be associated with cosmic ray induced extensive air showers (EAS), but for x-ray glows such an association has not been stated. In general, this issue touches on present discussions in the HEAP community, but is very far from the subject of our paper. The list of HEAP phenomena as a motivation for our study appears in section 1.1, and we now have extended this discussion in line 15: "Signals lasting longer than TGFs such as x- and gamma-ray glows or thunderstorm ground enhancements (TGEs) have also been observed near thunderclouds, from balloons, planes, or high mountains (McCarthy and Parks, 1985; Eack et al., 1996; Tsuchiya et al, 2007; Tsuchiya et al, 2008; Chilingarian et al, 2010; Chilingarian et al, 2011)."

Ad 3. We thank the referee for his review of available validation data from his TGE observations. But we repeat that code validation is not the purpose of our paper. Fortunately, his report will stay available for future use on the GMD-webpage together with our paper.

---

## Referee Comment (RC2) · A Chilingarian (Referee) · 9 Jul 2016

Second comment on the paper submitted to Geoscientific Model Development (GMD) "Evaluation of Monte Carlo tools for high energy atmospheric physics" by Casper Rutjes, David Sarria, Alexander B. Skeltved, Alejandro Luque, Gabriel Diniz, Nikolai Østgaard, and Ute Ebert (gmd-2016-147).

**A.Chilingarian, Yerevan Physics Institute, Armenia**

I want to comment following statement from the answer of Casper Rutjes et al. "Furthermore, long TGE events are supposed to be associated with cosmic ray induced extensive air showers (EAS), but for x-ray glows such an association has not been stated".

As we can see in Figure 1 (Figure 4, Chilingarian, 2014) there are different channels of generation of so-called secondary cosmic rays (SCR) measured my particles detectors located on the earth's surface or by balloons or aircraft in the terrestrial atmosphere.

---

## Author Comment (AC2) · 15 Jul 2016

Reply to: gmd-2016-147-RC2: 'Second comment on the paper submitted to Geoscientific Model Development (GMD) "Evaluation of Monte Carlo tools for high energy atmospheric physics" by Casper Rutjes, David Sarria, Alexander B. Skeltved, Alejandro Luque, Gabriel Diniz, Nikolai Østgaard', by Ashot Chilingarian, 09 Jul 2016

In which the referee comments on our statement – in our reply to gmd-2016-147-RC1 (not part of the paper): "Furthermore, long TGE events are supposed to be associated with cosmic ray induced extensive air showers (EAS), but for x-ray glows such an association has not been stated". ————————— The discussion is interesting, but it is out of the scope of the paper. The quoted natural phenomena just motivate our study.

We do not imply with the word association that TGE signal is actually an EAS signal.

We agree that the two are systematically different, as illustrated clearly in the referee's references and comment RC1 & RC2.

We want to say that the TGE signal must be caused by one or more energetic cosmic rays. In [Chilingarian et al, 2015] it is stated clearly: "furthermore lightning terminates TGE, did not gives rise to it". This in contrast to the x-ray glows, which could have a causality relation with the lightning stroke.

In addition, there seems to be a difference in terminology here. Dr. Chilingarian clearly states now and also illustrates in his Figure 1, that he uses the term "secondary cosmic particle" for any energetic particle in the atmosphere, independently of whether it was created by a cosmic ray or by radioactive decay or by runaway avalanches of thermal electrons in the electric fields of a thunderstorm. In contrast, we use "cosmic" only in relation with particles coming from outside the Earth's atmosphere.

---

## Referee Comment (RC3) · A Chilingarian (Referee) · 17 Jul 2016

Third comment on the paper submitted to Geoscientific Model Development (GMD) "Evaluation of Monte Carlo tools for high energy atmospheric physics" by Casper Rutjes, David Sarria, Alexander B. Skeltved, Alejandro Luque, Gabriel Diniz, Nikolai Østgaard, and Ute Ebert (gmd-2016-147). A.Chilingarian, Yerevan Physics Institute, Armenia "...there seems to be a difference in terminology here. Dr. Chilingarian clearly states now and also illustrates in his Figure 1, that he uses the term "secondary cosmic particle" for any energetic particle in the atmosphere, independently of whether it was created by a cosmic ray or by radioactive decay or by runaway avalanches of thermal electrons in the electric fields of a thunderstorm. In contrast, we use "cosmic" only in relation with particles coming from outside the Earth's atmosphere."

[Figure]

The new topic of high-energy atmospheric physics (HEAP) adopted knowledge from both atmospheric physics and high-energy astrophysics, and consequently groups of experts from both previously non-strongly overlapping communities. Therefore it is very important to use scientific terminology appropriately.

"particles coming from outside the Earth's atmosphere" in cosmic ray astrophysics communities called primary cosmic rays; and particles from cascades initiated by primary cosmic rays in interactions with terrestrial atmosphere called – secondary cosmic rays.

"The discussion is interesting, but it is out of the scope of the paper"

As I mention in my first comment the code verification problem 9topic of reviewed paper) is very important from technical point of view. However, it did not tell anything about how useful the code is for understanding nature of complicated HEAP problems. There exist thousands papers on simulations of particle cascades in atmosphere, but very few of them contain comparisons with experimentally measured parameters. My concern was that to firmly establish HEAP as new scientific discipline community needs multiple comparisons with existing experimental data to clarify physics of RREA cascades, seed particles, energy spectra of TGFs and TGEs, etc.

---

## Referee Comment (RC4) · Anonymous Referee #2 · 29 Aug 2016

This paper checks the physics used in several programs used to simulate terrestrial gamma-ray flashes by simulating simple cases (monoenergetic particles beams in air at 1 bar and 0 degrees C) and comparing the results. The relevant physics is throughly discussed and the reasons for differences between the simulation results are identified. This is highly useful since several of the codes were developed for other areas of physics and might have explicit or hidden deficiencies for high energy atmospheric physics. Obviously this falls short of a full validation against reality, e.g., some standard cross section used by all of the programs might deviate from the true value.

The results are clearly presented and explained . Issues are identified with several of the programs and advice is provided for future simulation efforts. The paper is a very useful contribution to high energy atmospheric physics modeling.

[Figure]

Specific Comments:

line 11, page 2: in the large-scale electric field theory, seed electrons might also originate from lightning leaders. See, e.g., section 4.6.2 of Dwyer, Smith and Cummer, Space Science Reviews, 2012.

page 5, line 25. In the discussion of positronium, the lifetimes are for vacuum. Will any positronium be disrupted in 1 bar air before decaying? Which of the programs, if any, handle two versus three photon decay of positronium? This effects the magnitude of the 511 keV line in TGF photon spectra, produced in the atmosphere.

technical corrections:

page 13, line 22: "dot" should be "not"

---

## Author Comment (AC3) · 29 Aug 2016

We agree that proper terminology is important, and we elaborated on it in our paper and in our last reply. We also discussed the questions of verification and validation and of available codes and the scope of our article already in our last reply.

---

## Author Comment (AC4) · 9 Sep 2016

We would like to thank the referee for his/her comments. We agree that further steps would be desirable in order to achieve full validation. The general purpose codes used in this benchmark (EGS5, FLUKA and Geant4) are in other physical contexts validated against experiments.

Two specific comments.

1. Referee #2 states: line 11, page 2: in the large-scale electric field theory, seed electrons might also originate from lightning leaders. See, e.g., section 4.6.2 of Dwyer, Smith and Cummer, Space Science Reviews, 2012.

Line 11, page 2 reads: The first electrons are typically supplied by cosmic particles

from the sun or from other galactic or extragalactic sources.

Or reply: We agree. We added the following line for more clarification: "High energy seed electrons might also origin from lightning leaders, from radioactive decay or from some mixed form of electron sources."

2. Referee #2 states: page 5, line 25. In the discussion of positronium, the lifetimes are for vacuum. Will any positronium be disrupted in 1 bar air before decaying? Which of the programs, if any, handle two versus three photon decay of positronium? This effects the magnitude of the 511 keV line in TGF photon spectra, produced in the atmosphere.

Line 25, page 5 reads: The standard implementation is that, when a positron drops below the low energy cut-off, it comes to rest immediately (in space and time). In reality the positron will come to rest over some distance and time, forming positronium (e.g. an e+e− bound state), before annihilation. The positronium has a lifetime depending on the spins of the positron and electron (Karshenboim, 2004), forming a singlet or triplet state with lifetimes of 124 ps or 139 ns (in vacuum), respectively.

Our reply: The positronium discussion was added to point out that reality is different than the standard implementation. We have added the following lines for more clarification; "If the triplet state is formed in a medium like air, the lifetime permits "pick-off" annihilation where an opposite spin electron from the medium will annihilate in singlet orientation before the triplet-oriented electron can collapse and annihilate with the positron, thus resulting in again 2 photons (instead of 3). Thus, besides a small time delay, the magnitude of 511 keV line in the photon spectrum is not changed. None of the codes with the settings used in this benchmark include positronium."

The typo has been corrected in the updated manuscript.

---

## Author Response (AR1)

**Authors response**

We would like to thank the referees for their constructive feedback. On the discussion page one can read our replies to their comments. Here we summarize the consecutive changes we have made in the manuscript. At the end of this document one may find the latex diff of the revised manuscript.

1. Referee #1 (Dr. A .Chilingarian) draws attention to Thunderstorm Ground Enhancements (TGEs) within the context of HEAP. We have added this in the introduction, first paragraph.
2. Referee #2 (anonymous) pointed out that seed electrons might also originate from lightning leaders. We have included the statement "High energy seed electrons might also origin from lightning leaders, from radioactive decay or from some mixed form of electron sources." in the fourth paragraph.
3. Referee #2 (anonymous) asks about life time and decay of ortho- or para-positronium in air. The requested information is added just above section 2.1.1. There we discuss as well that this physics is not included in any of the tested codes with the settings of our benchmark. We have indicated why it will not change the magnitude of the 511 keV line in the photon spectrum.
4. Referee #2 (anonymous) pointed out some typo which has been corrected.

Besides the corrections proposed by the referees we corrected one and added two references. We also corrected a few typos.

[revised manuscript text omitted]